# A Brazilian Rare-Disease Center’s Experience with Glucosylsphingosine (lyso-Gb1) in Patients with Gaucher Disease: Exploring a Novel Correlation with IgG Levels in Plasma and a Biomarker Measurement in CSF

**DOI:** 10.3390/ijms25052870

**Published:** 2024-03-01

**Authors:** Matheus Vernet Machado Bressan Wilke, Gabrielle Dineck Iop, Larissa Faqueti, Layzon Antonio Lemos da Silva, Francyne Kubaski, Fabiano O. Poswar, Kristiane Michelin-Tirelli, Dévora Randon, Wyllians Vendramini Borelli, Roberto Giugliani, Ida Vanessa D. Schwartz

**Affiliations:** 1Postgraduate Program in Medical Sciences, Universidade Federal do Rio Grande do Sul, UFRGS, Porto Alegre 90035003, Brazil; mtwilke@gmail.com (M.V.M.B.W.); rgiugliani@hcpa.edu.br (R.G.); 2Biodiscovery Laboratory, Hospital de Clínicas de Porto Alegre, Porto Alegre 90035903, Brazil; gabrielle.iop@gmail.com (G.D.I.); larissafaqueti@gmail.com (L.F.); layzon.antonio@gmail.com (L.A.L.d.S.); 3Biochemical Genetics Laboratory, Greenwood Genetics Center, Greenwood, SC 29646, USA; fkubaski@ggc.org; 4Medical Genetics Service, Hospital de Clínicas de Porto Alegre, Porto Alegre 90035003, Brazil; fposwar@hcpa.edu.br; 5Postgraduate Program in Genetics and Molecular Biology, Universidade Federal do Rio Grande do Sul (UFRGS), Porto Alegre 90035003, Brazil; 6LEIM-Biochemical Genetics Laboratory, Medical Genetics Service, Hospital de Clínicas de Porto Alegre, Porto Alegre 90035003, Brazil; ktirelli@hcpa.edu.br; 7BRAIN Laboratory, Hospital de Clínicas de Porto Alegre, Porto Alegre 90035003, Brazil; 8Neurology Service, Hospital de Clínicas de Porto Alegre, Porto Alegre 90035003, Brazil; 9Department of Genetics, Universidade Federal do Rio Grande do Sul (UFRGS), Porto Alegre 90035003, Brazil; 10Dasa Genômica, São Paulo 04078013, Brazil; 11Casa dos Raros, Porto Alegre 90035003, Brazil; 12Instituto Nacional de Doenças Raras—InRaras, Hospital de Clínicas de Porto Alegre, Porto Alegre 90035003, Brazil

**Keywords:** Gaucher disease, glucosylceramide, multiple myeloma, CSF

## Abstract

Gaucher disease (GD, OMIM 230800) is one of the most common lysosomal disorders, being caused by the deficient activity of the enzyme acid β-glucocerebrosidase (Gcase). Three clinical forms of Gaucher’s disease (GD) are classified based on neurological involvement. Type 1 (GD1) is non-neuronopathic, while types 2 (GD2) and 3 (GD3) are neuronopathic forms. Gcase catalyzes the conversion of glucosylceramide (GlcCer) into ceramide and glucose. As GlcCer accumulates in lysosomal macrophages, it undergoes deacylation to become glycosylsphingosine (lyso-Gb1), which has shown to be a useful and reliable biomarker for the diagnosis and monitoring of treated and untreated patients with GD. Multiple myeloma (MM) is one of the leading causes of cancer-related death among patients with GD and monoclonal gammopathy of undetermined significance (MGUS) is a non-neoplastic condition that can be a telltale sign of a B clonal proliferation caused by the chronic activation of B cells. This study aimed to quantify Lyso-Gb1 levels in dried blood spots (DBS) and cerebrospinal fluid (CSF) as biomarkers for Gaucher disease (GD) and discuss the association of this biomarker with other clinical parameters. This is a mixed-methods study incorporating both cross-sectional and longitudinal elements within a cohort design with a convenience-sampling strategy. Data collection took place from January 2012 to March 2023. Lyso-Gb1 extraction from DBS involved the use of a methanol–acetonitrile–water mixture, followed by incubation and centrifugation. Analysis was performed using UPLC-MS/MS with MassLynx software version 4.2 and the control group for the DBS measurements included general newborns. CSF Lyso-Gb1 was extracted using ethyl acetate, analyzed by UPLC-MS/MS with a calibration curve, and expressed in pmol/L. Lysosomal activity in CSF was assessed by measuring chitotriosidase (Cht), and other lysosomal enzyme activities were assessed as previously described in the literature. Patients with metachromatic leukodystrophy (MLD) were used as controls. Thirty-two treated patients (twenty-nine GD1 and three GD3, all on ERT except for one GD type on SRT with eliglustat) and three untreated patients (one GD1, one GD2, and one GD3) were included. When analyzing only the treated GD1 group, a significant correlation was found between lyso-Gb1 and age (rho = −0.447, *p* = 0.001), ChT, and IgG levels (rho = 0.73, *p* < 0.001; and rho = 0.36, *p* = 0.03, respectively). Five GD1 patients (three females, mean age 40 years) also had their CSF collected and analyzed. The average measurement of lyso-Gb1 in CSF was 94 pmol/L (range: 57.1–157.9 pmol/L) versus <6.2 pmol/L in the control group (MLD). This is the first time, to the best of our knowledge, that lyso-Gb1 has been associated with IgG levels. While this finding reflects a risk for MGUS or MM and not only chronic plasma B-cell activation, it still requires further studies. Moreover, the analysis of CSF lyso-Gb1 levels in GD1 patients was demonstrated to be significantly higher than the control group. This raises the hypothesis that CSF lyso-Gb1 may serve as a valuable indicator for neurological involvement in GD, providing insights into the potential implications for neurological manifestations in GD, including GD1. The correlation between lyso-Gb1 and ChT levels in treated GD1 patients further underscores the interconnectedness of lysosomal markers and their relevance in monitoring.

## 1. Introduction

Gaucher disease (GD, OMIM 230800) is one of the most common lysosomal disorders, with an estimated worldwide incidence of 50,000 to 100,000 live births [1]. Three clinical forms of GD are conventionally classified based on the neurological involvement: type 1 (GD1) is considered non-neuronopathic, whereas types 2 (GD2) and 3 (GD3) are considered the neuronopathic forms [2]. Even though GD1 is considered non-neuropathic, it has been shown that these patients are in a higher risk of developing Parkinson’s disease (PD) [3,4]. GD treatment includes enzyme-replacement therapy (ERT) and substrate-reduction therapy (SRT) that offers a significant improvement for most clinical parameters, with the exception of neurological impairment [5].

GD is caused by the deficient activity of the enzyme β-glucocerebrosidase (GCase; glucosylceramidase; EC 3.2.1.45) due to biallelic pathogenic variants in the *GBA1* gene. Gcase catalyzes the conversion of glucosylceramide (GlcCer) into ceramide and glucose. As GlcCer accumulates in lysosomal macrophages in the reticuloendothelial system, it undergoes deacetylation via an enzyme called acid ceramidase to become glycosylsphingosine (lyso-Gb1) [6].

The selection of biomarkers GD should take into consideration not only those that reflect the secondary inflammatory state of this monogenic disorder but also those that somehow indicate the primary pathogenesis. In the past, both (C-C motif) ligand 18 (CCL18), which indicates the overall Gaucher cell burden, as well as chitotriosidase (Cht), had been broadly used in the care of patients with GD [1,7]. Cht has been a known GD biomarker since 1994, and it remains the first and most commonly used biomarker in GD patients [8]. However, a major limitation of using this biomarker is that its activity increases in plasma during various inflammatory processes, and reduced/null activity can be observed due to null alleles in the gene that encodes it (*CHIT1*); and, this biomarker is also not specific to Gaucher disease [6,8]. Over the years, lyso-Gb1 has demonstrated its accessibility in clinical samples, quantifiability in an easy and reliable manner, and significant value both as a diagnostic tool and as an indicator of responses to therapeutic interventions [1,9]. Despite enzyme-replacement therapy (ERT), most GD2/3 patients, as well as some GD1 patients, have atypical manifestations such as lymphadenopathy, multiple myeloma, lymphoma, and neurotoxicity [1,5,10].

One of the challenges in the care of patients with GD is interpreting the values of biomarkers in various clinical contexts. In this report, we present the experience of a Rare Disease Center in using lyso-Gb1 values in a cohort of patients with GD. We also discuss the association of this biomarker with other clinical parameters, including a novel correlation with immunoglobulin G (IgG) values in plasma.

## 2. Results

Thirty-one treated patients (28 GD1, and 3 GD3, all on ERT except for one GD type on SRT with eliglustat) and three untreated patients (one GD1, one GD2, and one GD3) were included for the analysis of the levels of lyso-Gb1 in DBS, with a mean age of 38 years, ranging from 7 months to 71 years. The average duration of treatment in GD1 was 14 years (range = 4–27 years), while the average treatment duration in the GD3 group was 19 years (range = 16–23 years). Demographic and clinical findings from the treated patients are summarized in Table 1. Untreated patients are presented as separate cases in the Appendix A.

**(A)** 
**Lyso-Gb1 levels in dried blood spots showed a positive correlation with the treatment status and type of Gaucher disease**


A total of 57 lyso-Gb1 measurements were collected for this cohort during the study period, with 51 in the treated group, resulting in an average of 1.6 lyso-Gb1 measurements per patient (range = 1–4). In the treated group, lyso-Gb1 levels exhibited a significant reduction (median = 379 nmol/L, interquartile range [IQR] of 540.1) compared to untreated patients (median = 877 nmol/L, IQR of 958.5, *p* = 0.002, Rank-Biserial Correlation = −0.795 [95% CI: −0.918 to −0.532]) as shown in Figure 1A. The median lyso-Gb1 in the two control groups was 6.6 nmol/L (IQR of 2.6) in general newborns and 6.7 nmol/L (IQR of 3.23) in other LSDs. In the treated group, lyso-Gb1 levels were on median 112 nmol/L, with an IQR of 417.7 for the GD1 group, which was significantly lower (*p* < 0.001) than the treated GD3 group (median = 877 nmol/L, IQR 1247) as seen in Figure 1B.

**(B)** 
**Lyso-Gb1 levels correlated with IgG, age, and chitotriosidase in treated patients with Gaucher disease type 1**


When analyzing only the treated GD1 group, a significant correlation was found between lyso-Gb1 and age (rho = −0.447, *p* = 0.001), Cht, and IgG levels (*p* < 0.001 and *p* = 0.03, respectively) as seen in Figure 1C–E. For the same group, no significant differences were found between lyso-Gb1 values and severity scores, hb levels, and years of treatment, as shown in the heat map in Figure 2. In the GD3-treated group, these correlations were not made due to a smaller sample size and the unavailability of IgG levels in one patient.

**(C)** 
**Lyso-Gb1 Levels in cerebrospinal fluid were higher in GD1 patients than in controls**


Five GD1 patients (three females, mean age 40 years, pts 7,10, 13, 16, and 19) also had their CSF collected and analyzed. The average measurement of lyso-Gb1 in CSF was 94 pmol/L (range: 57.1–157.9 pmol/L) as shown in Table 2. In the control group (MLD), the average measurement was <6.2 pmol/L.

**(D)** 
**Lyso-Gb1 levels followed the trend of chitotriosidase decrease after treatment in treatment-naïve patients with Gaucher disease**


The Lyso-Gb1 measurements were also evaluated in three GD treatment-naive patients who had recently started ERT (one with GD1, one with GD2, and one with GD3) after being referred to the Gaucher Disease Referral Center from their local primary-care facilities. In two of the patients, we observed the expected decrease in lyso-Gb1 levels after the initiation of treatment as shown in Appendix A.

The patient with GD2 showed an increase in both lyso-Gb1 and Cht measurements after the beginning of ERT (Appendix A). Unfortunately, the patient with GD2 passed away due to sepsis of unclear etiology, and the development of Hemophagocytic Lymphohistiocytosis Syndrome (HLH). As a result, we were unable to further evaluate the response to treatment in this case. The complete clinical cases can be found in the Appendix A.

## 3. Discussion

The care of patients with GD relies on validated biomarkers that aid the diagnosis, prognosis, follow-up, treatment decisions, and pathophysiologic understanding of this disease [1]. The present study reports the experience of a Brazilian Rare-Disease Center’s experience of the use of lyso-Gb1 in the care of patients with GD.

The finding in our cohort of a significant correlation between the values of lyso-Gb1, and ChT has already been extensively reported in the literature as has the decrease in this biomarker after the beginning of GD-specific treatments such as ERT and SRT [1,11]. The inverse relationship between this biomarker and patient’s age was also reported previously with the rationale that children have more severe disease at presentation than adults, and therefore increased lyso-gb1 values [12]. According to an interesting paper that includes data from 22 pediatric patients with GD, lyso-Gb1 levels generally increased with age also among treatment-naïve individuals; however, the majority of treatment-naïve individuals had lyso-GL1 levels under 40 ng/mL [13]. In our cohort, we were only able to assess the inverted correlation of Lyso-Gb1 values and age in GD1-treated patients, as shown in Figure 1C. Interestingly, age and disease severity, represented by the Disease Severity Score (DS3) and Zimran Severity Score Index (SSI), had a direct statistically significant correlation as shown in Figure 2, even though disease severity and Lyso-Gb1 measurements did not show an association in the present study. Our findings also did not reveal any association between the levels of Lyso-Gb1 and the duration of treatment, as illustrated in Figure 2. We hypothesize that this lack of association might be attributed not only to a small sample size but also to some patients not adhering to treatment, as shown in Table 1. This is supported by a stronger association between lyso-Gb1 and therapeutic interventions in GD compared to the association between lyso-Gb1 and age in the literature [1].

A threshold of 12 ng/mL (25.99 nmol/mL) has been used for plasma lyso-Gb1 to differentiate patients with GD from healthy probands, and from patients with other lysosomal storage diseases with 100% sensitivity and 100% specificity [1,14]. The increase in lyso-Gb1 while receiving ERT, as noticed in the GD2 patient in the Appendix A, was reported in eight children as being associated with compliance and significant weight gain (>15%) [15]. We postulate that the worsening of the patient’s clinical picture, including a suspicion of HLH, could be the reason for the elevation of this biomarker as no weight gain was reported [16]. The hypothesis that GD 2 could possibly trigger HLH has already been reported in the literature as showing a poor response to HLH treatment in these patients [17]. Due to the paucity of reports in the literature and their variability in longitudinal measurements, no specific cutoff for lyso-Gb1 levels has been established as a sole indication for the beginning of ERT or to the response to treatment [15]. Therefore, when interpreting lyso-Gb1 concentrations among different studies, the focus should be on trends (whether they are decreasing or increasing) rather than on absolute quantification, as demonstrated in Appendix A.

The present study also reported the measurement of lyso-Gb1 in the CSF of five patients with GD1 presentation, suggesting an opportunity to better understand the role of this biomarker in, for example, the relationship between GD1 and PD. One of the hypotheses for this association is that the dysfunction of the autophagic-lysosomal pathway caused by the mutated GCase would lead to aberrant lipid metabolism. This may involve the accumulation of glycosphingolipids, glucosylceramide, and glucosylsphingosine, ultimately leading to the further accumulation of alpha-synuclein [18]. The same patients were previously assessed for non-motor symptoms of PD in a separate study conducted by our group [3]. Patients 7, 13, and 19 exhibited at least one non-motor symptom (NMS), but interestingly, they were not the patients with the highest levels of lyso-Gb1 in the CSF, as indicated in Table 2. We hypothesize that CSF measurement of lyso-Gb1 could serve as a predictor for the development of PD in GD1 patients; however, further studies are required to substantiate this claim. The ERT is thought not to cross the blood–brain barrier (BBB); therefore, a combination of ERT plus ambroxol (as a chemical chaperone) has been shown to reduce lyso-Gb1 concentration in CSF by 26% versus the baseline in a multicenter open-label pilot study of five patients with neuronopathic GD [19]. The baseline values for this cohort included four patients with GD3 and one patient with GD2, showing a range of 174 pg/mL (18–635). Interestingly, there is an overlap in measurements when compared to the reported values for the five patients with GD1, considered the non-neuronopathic form, reported in our study, which were 94 pmol/L (range: 57.1–157.9 pmol/L). Lyso-Gb1 has been reported as being highly abundant in the brain tissue of patients with GD2 and GD3 and not so much in GD1 with the exception of one severe case, published as GD1 due to the absence of neurological symptoms, that passed away at the age of three due to cardiac failure [1,20]. In the anatomopathological report, it was mentioned that the levels of lyso-Gb1 of this severe patient were compared to GD3 levels. In contrast, the levels in the brain of a 56-year-old male individual with GD1 were very low [20]. The same MLD group was also tested for ChT levels in CSF in our previous study, compared to our GD1 patients where CSF was available, demonstrating that ChT levels exhibited a higher increase in the MLD group compared to the GD group, although they were not statistically different. Interestingly, lyso-Gb1 levels were found to be higher in the present study in the GD1 group, which corroborates the specificity of this biomarker in GD [3]. Animal studies have demonstrated the correlation between the increase in lyso-Gb1 and alpha-synuclein (α-syn, the hallmark of PD) accumulation [21]. In a study using conduritol-β-epoxide (CBE), a potent irreversible competitive inhibitor of GCase, it was demonstrated in mice that insoluble aggregates of α-syn accumulate in the substantia nigra, along with altered levels of GluCer and lyso-Gb1 after 28 days of enzymatic inhibition [22].

Multiple myeloma (MM) is one of the leading causes of cancer-related death among patients with GD with a relative risk of 37.5-fold [23]. The most common type of heavy chain produced in myeloma is IgG, followed by IgA and then IgD [24]. The monoclonal gammopathy of undetermined significance (MGUS) is a noncancerous condition that can be a telltale sign of a B clonal proliferation caused by the chronic activation of B cells. Due to the increased frequency of these conditions in the GD population, the routine measurement of immunoglobulins has been made part of the care of GD patients [25,26]. In the Brazilian clinical-practice guidelines for GD, this monitoring is performed every 3–5 years [25]. In the present study, IgG levels were significantly correlated with lyso-Gb1 levels in patients with GD1. Hypergammaglobulinemias, also known as polyclonal gammopathy (PG), can occur in many clinical contexts, such as chronic inflammation and liver diseases, and it is caused by an overproduction of immunoglobulins via plasma cells or B-lymphocytes [26,27,28]. Interestingly, the direct role of lyso-Gb1 in the chronic activation of B cells was already demonstrated in animal models for GD: the injection of lyso-Gb1 led to a further increase in Fas+GL7+ splenic germinal-center B cells as well as an increase in bone marrow CD38+CD138+ plasma cells [29]. Treatment with eliglustat also showed a reduction in both anti-lyso-Gb1 antibodies and clonal immunoglobulins in *GBA1*−/− animals [29]. In a cohort of 68 GD1 patients, aiming to explore the utility of gammopathy screening, the overall risk of MM in GD is 5.9–51.1 times higher than expected in these patients [10]. This study also shows the possible therapeutic benefit of ERT/SRT in reducing clonal immunoglobulins, with data also suggesting a possible remission of MGUS with ERT/SRT [10]. Our findings allow us to further hypothesize that hypergammaglobulinemia is observed in GD, not only due to the chronic inflammatory state of these patients, but also due to the chronic antigenic stimulation of the B lymphocytes caused by lyso-Gb1. However, this still needs to be further elucidated. Our study is exploratory in nature and further studies with larger samples should be conducted to perform more complex analyses to confirm the correlation between lyso-Gb1 and IgG levels and the direct correlation of hypergammaglobulinemia and the development of MM or MGUS. While, in one paper, high levels of IgG (>1725 mg/dL) and age were correlated to the development of neoplasia, in another, the serum levels of free light chains of immunoglobulins were not found to be predictive for the development of neither MGUS or MM [26,27]. Therefore, further studies are necessary to better understand the role of lyso-Gb1 also as a biomarker for MGUS and MM.

## 4. Materials and Methods

### 4.1. Ethics

A mixed-methods study was conducted, incorporating both cross-sectional and longitudinal elements within a cohort design. A convenience-sampling strategy was employed for data collection, as the study encompassed the SARS-CoV-2 pandemic, and many in-person appointments were cancelled during this period. The collection of lyso-Gb1 in dried blood spots took place from January 2018 to March 2023, and in cerebrospinal fluid (CSF), it occurred from January 2021 to November 2022. The study protocol (ID-2020-0358) was approved by the Ethics Committee of Hospital de Clínicas de Porto Alegre (HCPA), Brazil, and all patients provided written informed consent. Patients who had biochemical and genetic diagnoses of GD and were followed at the Gaucher Disease Referral Center at HCPA were eligible for inclusion in the study. All patients received standard clinical management as established in the Brazilian clinical-practice guidelines for GD, which included the measurement of Cht activity, complete blood counts, and a medical visit every three months [25]. In addition to the standard clinical management, lyso-Gb1 measurements were also assessed in these patients, as described below. All available lyso-Gb1 measurements within our cohort were included in our study.

### 4.2. Lyso-Gb1 Quantification in Dried Blood Spots

Lyso-Gb1 was prepared as described by Polo et al., 2019 [30]. In summary, Lyso-Gb1 was extracted from a 3.2 mm diameter dried blood spot (DBS) with 100 µL of a mixture of methanol–acetonitrile–water (80:15:5, *v*/*v*), containing the internal stable isotope standard, and incubated for 1 h at 45 °C with orbital shaking (500 rpm). After the incubation, samples were centrifuged for 5 min at 4500 rpm. The supernatant was transferred to a 96-well plate and 50 µL of ultrapure water was added [30].

Lyso-Gb1 measurements were performed on a Waters Xevo TQ-S micro (Waters Co., Manchester, UK) mass spectrometer. Data acquisition and processing were accomplished with MassLynx 4.2 software, and by TargetLynx™ Application Manager, supplied by Waters Technologies. Chromatographic separation was achieved on an XSelect^®^ CSH™ C18 column (2.1 × 50 mm; 3.5 µm), maintained at 55 °C. The mobile phase consisted of a 3.5 min gradient system combining 70:30 ultrapure water/acetonitrile with 0.1% formic acid (A) to 65:35 isopropanol/acetonitrile with 0.1% formic acid (B), at a flow rate of 0.8 mL/min. This was as follows: 0–0.75 min, 99.5–75% A; 0.75–1 min, 75–40% A; 1–1.5 min, 40–25% A; 1.5–1.8 min, 25–0% A; 1.8–2.8 min, 0% A; and 2.8–2.85 min, 0–99.5% A, with additional 45 s to re-equilibrate the column. The injection volume was 10 μL.

Mass spectrometry was set in the positive ion mode using an electrospray ionization source (ESI), with multiple reaction monitoring (MRM). The two MRM transitions (*m*/*z*) selected for lyso-Gb1 and lyso-Gb1-d5 were 462.3–282.3 and 467.3–287.3, respectively, with 30 V for cone and 20 V for collision. The levels of lyso-Gb1 were expressed as nmoL/L. All samples were measured from 2020 to 2023 (after the method was developed and validated). The method has not been modified since its validation in 2020.

Two different control groups (general newborns, and other lysosomal-storage disorders (LSDs): 2 patients with metachromatic leukodystrophy (MLD), 2 patients with mucopolysaccharidosis (MPS) type IIIB, 1 patient with MPS type IIIC, 2 patients with MPS type IVA, 1 patient with MPS type VI, and 1 patient with MPS type MPS VII, for whom which DBS was available to test, were also tested.

### 4.3. Cerebrospinal Fluid Collection

CSF samples were obtained from the patients previously described by Wilke et al. (2023) in a study designed to assess non-motor symptoms (NMS) in adult patients with GD1. Details regarding the CSF-collection methodology can be found in the original study [3].

### 4.4. Lyso-Gb1 Quantification in Cerebrospinal Fluid

Lyso-Gb1 was extracted from CSF samples (400 µL) by adding 1000 μL of ethyl acetate and 10 µL of 100 nmol/L of LysoGb1-d5 in water/acetonitrile (8:2 *v*/*v*). Samples were homogenized in vortex for 30 s, and centrifuged at 13,000 rpm for 10 min. The supernatant was transferred to an eppendorf tube, and the extraction process was repeated with 1000 μL of ethyl acetate. The ethyl acetate was removed from the supernatant under reduced pressure (SpeedVac DNA 130 system, ThermoFisher, Asheville, NC, USA) and the sample was reconstituted using 100 µL of a mixture of water/acetonitrile (8:2 *v*/*v*) with 0.1% of formic acid.

Lyso-Gb1 determination in CSF was calculated using a calibration curve (ranging from 6.25 to 162.5 pmol/L). CSF calibration curve was constructed using the linear least square regression analysis, based on the ratio between the chromatogram peak areas of lyso-Gb1 and the lyso-Gb1-d5 (IS) versus lyso-Gb1 concentration. Slopes, intercepts, and coefficients of determination (R^2^) were determined. UPLC-MS/MS analyses were performed similarly according to the procedure previously described in Section 4.2, with some modifications in the chromatographic conditions which included using a Waters Acquity BEH C18 column (2.1 × 50 mm; 1.7 µm) and changes in the composition of the gradient system, with 0.1% aqueous formic acid (A) and acetonitrile with 0.1% formic acid (B), at a flow rate of 0.5 mL/min. The levels of lyso-Gb1 were expressed as pmoL/L. A control group consisting of 6 patients with MLD was employed due to the availability of CSF aliquots from a separate study conducted concurrently.

### 4.5. Lysosomal Enzymes’ Activity in Cerebrospinal Fluid

The Cht activity was measured using a fluorimetric method using artificial 4-MUβ-D-triacetyl chitotriosidase substrate, following the protocol previously described [31]. Additionally, we assessed the activity of other lysosomal enzymes, including α-mannosidase, β-glucuronidase, and β-hexosaminidases in these samples, as published in a complementary study [3].

### 4.6. Statistics

The Mann–Whitney test was employed to examine differences in medians between treated and untreated individuals for lyso-Gb1. Additionally, the same test was conducted to assess the disparity between those treated with GD1 and GD3 in terms of lyso-Gb1 values. To validate the assumptions of the *t*-test, the Shapiro–Wilk normality test was conducted, along with the Levene test to verify variance equality. Effect size was determined using the Rank-Biserial correlation.

To ascertain whether any correlation existed between the lyso-Gb1 of each patient and their contemporary clinical and laboratory data (age, time of treatment, Cht activity, hemoglobin, platelet count, IgG levels and GD severity scores), the Spearman correlation analyses were conducted. These analyses were segregated based on GD1 and GD3. *p*-values below 0.05 were considered statistically significant. Statistical analysis was performed using JASP version 0.16.2.0 (JASP Team, 2022) [32].

## 5. Conclusions

This paper reports the experience of a Brazilian Rare-Disease Center in the use of lyso-Gb1 as a biomarker for patients with GD. This is the first time, to the best of our knowledge, that lyso-Gb1 has been positively correlated with IgG levels and that GD1 patients are shown to have elevated CSF lyso-Gb1 despite an absence of neuronal symptoms. While this finding reflects a risk for MM and PD, it still requires further studies.

## Figures and Tables

**Figure 1 ijms-25-02870-f001:**
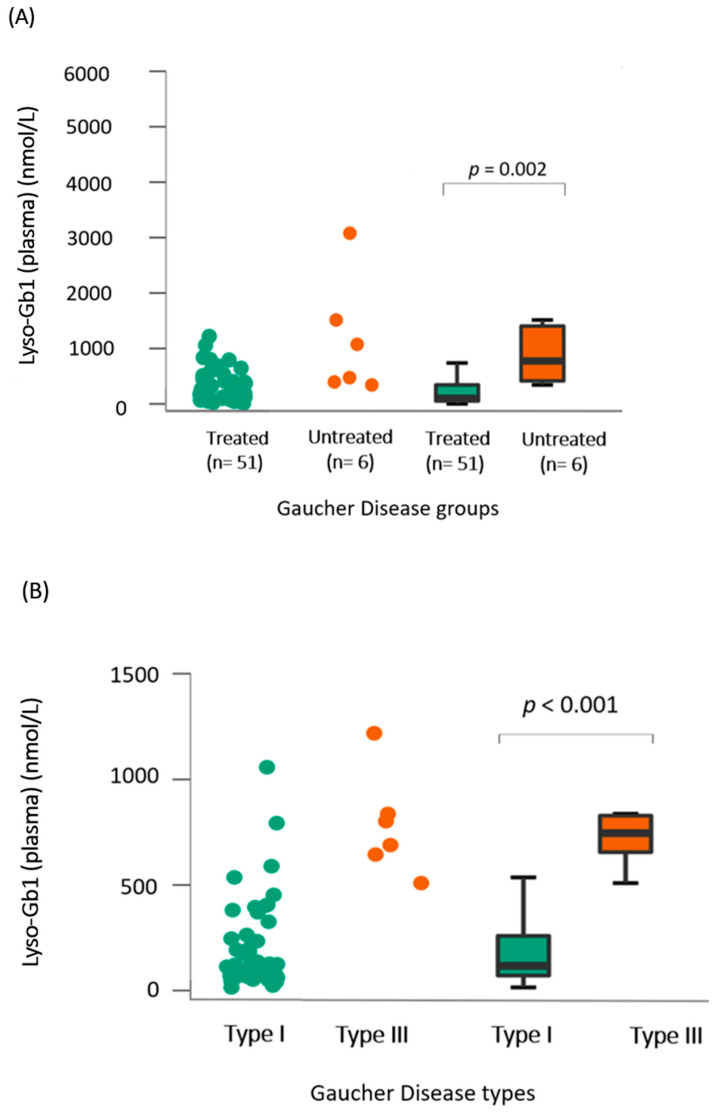
Lyso-Gb1 dried blood-spot analysis in our cohort (**A**) Comparison between lyso-Gb1 levels in treated (green) and untreated (orange) patients showed a significant difference with *p* = 0.002. (**B**) Comparison between lyso-Gb1 levels in treated GD1 (green) and GD3 (orange) showed a significant difference with *p* < 0.001. (**C**–**E**) Significant correlation analysis between lyso-Gb1 levels in treated GD1 patients and their age, IgG levels, and chitotriosidase levels in plasma, yielding *p*-values of 0.001, 0.03, and <0.001, respectively.

**Figure 2 ijms-25-02870-f002:**
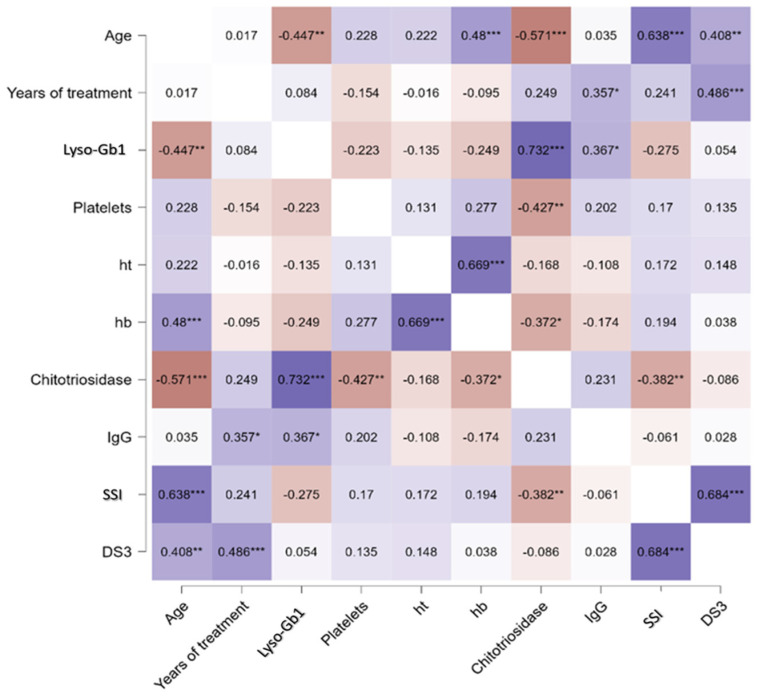
Heat map of the Spearman correlation between lyso-Gb1 and other clinical variables. Correlations were performed between each blood biomarker for treated GD1. *Ht*, hematocrit, *hb*, hemoglobin, *SSI* Zimran Severity Score Index, *DS3* Disease Severity Score. * *p* < 0.05, ** *p* < 0.01, *** *p* < 0.001. Colors represent direct correlation (purple) and inverse correlations (red), with different shades representing the strength of these associations.

**Table 1 ijms-25-02870-t001:** Demographics and latest clinical characteristics of treated patients with Gaucher disease type 1 (n = 29) and type 3 (n = 3).

Patient	Gaucher Type	Gender	Age (Years)	Genotype (NM_000157.3)	Treatment Duration (Years)	Current Treatment	Adherence to Treatment *	Severity Scores		Platelet (/mcL)	Hb (g/dL)	Lyso-Gb1 DBS (ng/mL)	ChT Plasma (nmol/h/mL)	IgG Plasma (mg/dL)
								SSI	DS3					
1	I	F	*12*	[c.(1226A>G)];[c. (721G>A)] [p.(Asn409Ser)]; [p.(Gly241Arg)]	4	ERT	Y	0	0	266,000	13.8	199	1724	795
2	I	M	*26*	[c.(1226A>G)];[c. (721G>A)] [p.(Asn409Ser)]; [p.(Gly241Arg)]	18	ERT	Y	0	0.66	171,000	14.9	234.1	4951	1073
3	I	F	*27*	[c.(1226A>G)];(RecNciI) [p. (Asn409Ser)];(RecNciI)	19	ERT	N	0	0.16	239,000	12.5	76.7	400	1670
4	I	F	*27*	[c.(1226A>G)];[c. (1448T>C)] [p. (Asn409Ser)];[p. (Leu483Pro)]	6	ERT	Y	1	0	225,000	12.5	127	3095	1539
5	I	F	*29*	[c.(1226A>G)];[c. (1448T>C)] [p. (Asn409Ser)];[p. (Leu483Pro)]	13	ERT	Y	1	0	166,000	11.5	61.6	425	1018
6	I	F	*29*	[c.(1226A>G)];[c.(1226A>)] [p.(Asn409Ser)];[p.(Asn409Ser)]	20	ERT	N	1	3.75	154,000	12.2	793	17,979	1463
**7**	**I**	**M**	** *30* **	**[c.(1226A>G)];[c c.1388+1G>A)]** **[p.(Asn409Ser)];[p. ?]**	**19**	**ERT**	**Y**	**3**	**1.2**	**149,000**	**15.4**	**453**	**1988**	**1018**
8	I	M	*32*	[c.(1226A>G)];[c.(1448T>G)] [p.(Asn409Ser)];[p.(Leu483Arg)]	27	ERT	NA	1	2.5	157,000	13.3	263.4	15,917	1463
9	I	F	*32*	[c.(1226A>G)];(RecNciI) [p. (Asn409Ser)];(RecNciI)	22	SRT	Y	0	2.5	146,000	11.6	371	3699	1098
**10**	**I**	**M**	** *35* **	**[c.(1226A>G)];(RecNciI) [p. (Asn409Ser)];(RecNciI)**	**21**	**ERT**	**Y**	**3**	**1**	**212,000**	**16.1**	**77**	**1243**	**NA**
11	I	F	*37*	[c.(1226A>G)];[c. (1448T>C)] [p. (Asn409Ser)];[p. (Leu483Pro)]	19	ERT	Y	0	0	182,000	13.3	192.7	2306	NA
12	I	F	*40*	[c.(1226A>G)];(RecNciI) [p. (Asn409Ser)];(RecNciI)	27	ERT	N	NA	NA	141,000	NA	1057.7	8657	1255
**13**	**I**	**F**	** *41* **	**[c.(1226A>G)];[c. (1448T>C)]** **[p. (Asn409Ser)];[p. (Leu483Pro)]**	**13**	**ERT**	**N**	**9**	**2.9**	**148,000**	**10.7**	**71.6**	**7464**	**1923**
14	I	M	*42*	[c.(1226A>G)];[c. (1448T>C)] [p. (Asn409Ser)];[p. (Leu483Pro)]	11	ERT	Y	0	0	142,000	14.4	112	1194	2020
15	I	F	*43*	[c.(1226A>G)];(RecNciI) [p. (Asn409Ser)];(RecNciI)	7	ERT	Y	2	1	221,000	13	589	736	954
**16**	**I**	**F**	** *44* **	**[c.(1226A>G)];[c. (1448T>C)]** **[p. (Asn409Ser)];[p. (Leu483Pro)]**	**14**	**ERT**	**Y**	**1**	**0.7**	**171,000**	**15**	**122**	**3022**	**1078**
17	I	F	*48*	[c.(1226A>G)];[c.(1226A>G)] [p.(Asn409Ser)];[p.(Asn409Ser)]	1	ERT	NA	NA	NA	NA	NA	246	NA	NA
18	I	M	*49*	[c.(1226A>G)];[c. (1448T>C)] [p. (Asn409Ser)];[p. (Leu483Pro)]	10	ERT	Y	5	0.4	199,000	14.4	15	116	NA
**19**	**I**	**F**	** *53* **	**[c.(1162G>A)];[c.(1214G>A)]** **[p.(Glu388Lys)];[p.(Ser405Asn)]**	**10**	**ERT**	**Y**	**3**	**1.8**	**285,000**	**14.7**	**23.57**	**643**	**885**
20	I	F	*55*	[c.(1226A>G)];(RecNciI) [p. (Asn409Ser)];(RecNciI)	20	ERT	Y	5.7	13	405,000	14.7	113	1216	1700
21	I	F	*56*	[c.(1162G>A)];[c.(1214G>A)] [p.(Glu388Lys)];[p.(Ser405Asn)]	8	ERT	Y	4	0.4	248,000	13,4	98	492	1202
22	I	M	*57*	[c.(1226A>G)];(RecNciI)] [p. (Asn409Ser)];(RecNciI)]	11	ERT	Y	2	0.5	141,000	14.9	183.5	965	NA
23	I	M	*61*	[c.(1226A>G)];(RecNciI)] [p. (Asn409Ser)];(RecNciI)]	10	ERT	Y	3	0	160,000	16.1	64.56	1170	1209
24	I	F	*66*	[c.(1226A>G)];(RecNciI)] [p. (Asn409Ser)];(RecNciI)]	24	ERT	Y	23 ^¥^	1	127,000	12.8	76	1741	783
25	I	F	*67*	[c.(1226A>G)];(RecNciI)] [p. (Asn409Ser)];(RecNciI)]	10	ERT	Y	4	4.5	143,000	14.8	52	146	876
26	I	M	*67*	[c.(1226A>G)];(RecNciI)] [p. (Asn409Ser)];(RecNciI)]	14	ERT	Y	8	2.1	213,000	14.5	71.58	401	1816
27	I	M	*68*	[c.(1226A>G)];[c.(1226A>G)] [p.(Asn409Ser)];[p.(Asn409Ser)]	12	ERT	Y	0	0.5	160,000	14.2	85.86	1529	1573
28	I	F	*71*	[c.(1226A>G)];[c.(1448T>G)] [p.(Asn409Ser)];[p.(Leu483Arg)]	11	ERT	Y	10	4.5	309,000	13.9	120.57	257	1116
29	III	M	*25*	[c. (1448T>C)]; [c. (1448T>C)] [p. (Leu483Pro)];[ p. (Leu483Pro)]	23	ERT	Y	20 ^¥^	0	184,000	15.7	837	4365	1561
30	III	M	*30*	[c. (1448T>C)]; [c. (1448T>C)] [p. (Leu483Pro)];[ p. (Leu483Pro)]	16	ERT	Y	NA	19.5	586,000	12.5	802	11,801	NA
31	III	F	*33*	[c. (1448T>C)]; [c. (1448T>C)] [p. (Leu483Pro)];[ p. (Leu483Pro)]	18	ERT	Y	25 ^¥^	0.6	401,000	12.4	689	12,560	1458

*ERT* enzyme-replacement therapy, *SRT* substrate-reduction therapy, *SSI* Zimran Severity Score Index (mild = 0–10; moderate = 11–19; severe ≥ 20), *DS3* Disease Severity Score (mild ≤ 3.00; moderate = 3.00–5.99; marked = 6.00–19), Hb hemoglobin, *ChT activity* chitotriosidase activity (normal range: 78.5 nmol/h/mL), *CSF* cerebrospinal fluid, *NA* Not available. *DBS* dried blood spots. * Adherence to treatment was defined as >50% of the treatment performed as prescribed. ^¥^ Higher SSI score due to neurological symptoms. In bold, patients with CSF collected.

**Table 2 ijms-25-02870-t002:** Biomarkers and lysosome enzymes measurements on cerebrospinal fluid of patients with Gaucher disease type 1 (n = 5).

Patient	Lyso-Gb1 (pmol/L)	Cht (nmol/h/mL)	β-hexo A (nmol/h/mL)	β-hexo B (nmol/h/mL)	Total β-hexo (nmol/h/mL)	% β-hexo A	β-glu (nmol/h/mL)	α-man (nmol/h/mL)	NMD (n)
7	109.04	102	104	3	107	97	3.3	2.2	2
10	83.49	97	81	107	188	43	5.5	0.8	1
13	57.13	19	105	15	120	88	7	0.9	2
16	154.87	81	190	59	249	76	8.5	4.7	0
19	66.43	17	36	28	65	56	5	0.25	3

Cht chitotriosidase (reference range in plasma = 8.8–132 nmol/h/mL), β-hexo β-hexosaminidase (reference range in plasma = 265–1219 nmol/h/mL), β-glu β-glucuronidase (reference range in plasma = 30–300 nmol/h/mL), α-man α-mannosidase (reference range in plasma = 17–56 nmol/h/mL). NMD = non motor symptoms of Parkinson’s disease. The lysosome enzymes measurements and NMD were already previously published [3].

## Data Availability

If necessary, further information is available from the corresponding author on reasonable request.

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
