# Peer review of "A Brazilian Rare-Disease Center’s Experience with Glucosylsphingosine (lyso-Gb1) in Patients with Gaucher Disease: Exploring a Novel Correlation with IgG Levels in Plasma and a Biomarker Measurement in CSF"

_ijms, 2024, doi:10.3390/ijms25052870_

Round 1

Reviewer 1 Report

Comments and Suggestions for Authors

Thank you for the opportunity to review the manuscript entitled "A Brazilian Rare Disease Center's Experience with Glucosylsphingosine (lyso-Gb1) in Patients with Gaucher Disease: Novel Correlation with IgG Levels and Biomarker Measurement in CSF." Here are my comments and suggestions:

  1. General comment-

  2. The  lyso-Gb1 levels in the CSF and the correlation of plasma lyso-Gb1 with IgG are interesting. However, these novel findings seem overshadowed by information that is already well-established in the field. I suggest revising the paper to emphasize these unique insights.

  3.  
  4. Study Design and Methodology:

    • The manuscript describes a cross-sectional study spanning 11 years (2012-2023). It is unclear whether the method of lyso-Gb1 measurement remained consistent throughout these years. Clarification on this would strengthen the methodology section.
    • When multiple samples were taken from the same individual over the years, the criteria for selecting which sample was analyzed (first, last, or randomly chosen) is not specified. Additionally, the study seems to include longitudinal elements for GD3 and some GD1 cases, which contradicts the cross-sectional approach. This needs clarification.
  5. Statistical Analysis:

    • The manuscript inaccurately describes the use of the Spearman and Mann–Whitney U tests. The Spearman test, being non-parametric, is not appropriate for parametric variables, as stated. Similarly, the Mann–Whitney U test compares differences between groups, not correlations. This section requires correction for methodological accuracy.
  6. Results :

    • - Table 1 appears overly detailed for the main text. I suggest moving this detailed table to the supplementary material and presenting aggregated data in the main text. Additionally, including IgG levels in the detailed table would be beneficial.
    • - If the distribution of lyso-Gb1 is not normal, using median values instead of averages would be more appropriate.
    • - In comparing the treated GD1 and GD3 groups, it's important to consider whether the treatment durations were similar, as this could impact lyso-Gb1 levels.
    • The significant correlation found between lyso-Gb1 and age in the treated GD1 group warrants further analysis. Adjusting for variables like disease severity, treatment duration, adherence to treatment, etc, is needed to validate this finding. A similar adjustment is necessary for understanding the relationship between IgG levels and lyso-Gb1.
    • The absence of correlations in the GD3 treated group could be due to the small sample size (only three cases). This limitation should be acknowledged.
  7. Section E:

    • Section E does not contribute significantly to the study of the existing literature. I recommend considering its removal for a more focused and impactful paper.
  8. Discussion:

  9. - "We believe that the CSF measurement of lyso-Gb1 could be used as a predictor of the development of PD in GD1 patients."- what is the basis of this belief? Is it something that can be learned from your data? The samples were taken from a PD prodromal study- was there any association between the lyso-Gb1 levels and prodromal findings?

  10.  
  11. Minor Points:

    • The gene name should be updated to GBA1 as per recent nomenclature changes.
    •  
    • References 8 and 9 are duplicated; this should be corrected.
    •  

Comments on the Quality of English Language

I suggest using English Medical Writing to improve the readability of the study. 

Author Response

General comment-

  1. The lyso-Gb1 levels in the CSF and the correlation of plasma lyso-Gb1 with IgG are interesting. However, these novel findings seem overshadowed by information that is already well-established in the field. I suggest revising the paper to emphasize these unique insights.

Thank you for your comments. We've removed the previous "Section E" from the manuscript and relocated it to the supplementary materials. Furthermore, we've repositioned section D, which discusses the findings on CSF levels, to Section C to showcase this intriguing information earlier in the results. You can find these details in lines 227-232. Throughout the manuscript, we emphasize that case descriptions are available in the supplementary material (lines 239, 241, 244 and 288). We believe retaining established information (such as the association with treatment and Gaucher Disease type) in the field enhances our manuscript, detailing our center's experience with lyso-Gb1 and confirming findings consistent with published literature.

  1. Study Design and Methodology:

The manuscript describes a cross-sectional study spanning 11 years (2012-2023). It is unclear whether the method of lyso-Gb1 measurement remained consistent throughout these years. Clarification on this would strengthen the methodology section.

Answer: Thank you for your question. We utilized a method developed and published by Polo et al., 2019, for the analysis of Lyso-Gb1 in dried blood spots (DBS). Additionally, we established an in-house method for the quantification in cerebrospinal fluid (CSF). All samples were stored at -20°C, and measurements were conducted from 2020 to 2023, following the development and validation of the method. It's essential to note that the method has remained unchanged since its validation in 2020. This information has been incorporated into sentences 137-139.

  1. When multiple samples were taken from the same individual over the years, the criteria for selecting which sample was analyzed (first, last, or randomly chosen) is not specified. Additionally, the study seems to include longitudinal elements for GD3 and some GD1 cases, which contradicts the cross-sectional approach. This needs clarification.

Answer: Thank you for your comment. Indeed, there are longitudinal aspects in our cohort study that don't fit into the category of a transversal study. This has been addressed with the statement, "A mixed-methods study was conducted, incorporating both cross-sectional and longitudinal elements within a cohort design. A convenience sampling strategy was employed for data collection," as seen in lines 39-40 and 103-105. All available lyso-Gb1 measurements within our cohort were incorporated into our study, as mentioned now in lines 115-116. Some patients had only 1 measurement, while others had up to 4. This accounts for an average of 1.6 lyso-Gb1 measurements per patient (range 1-4), as stated in lines 209-210.

  1. The manuscript inaccurately describes the use of the Spearman and Mann–Whitney U tests. The Spearman test, being non-parametric, is not appropriate for parametric variables, as stated. Similarly, the Mann–Whitney U test compares differences between groups, not correlations. This section requires correction for methodological accuracy.

Answer: Thank you for your comments and for bringing this to our attention. The statistical section was modified to correctly reflect the tests used: “The Mann-Whitney test was employed to examine differences in medians between treated and untreated individuals for lyso- Gb1. Additionally, the same test was conducted to assess the disparity between those treated with Type I and Type III for lyso-Gb1. To validate the assumptions of the t-test, the Shapiro-Wilk normality test was conducted, along with the Levene test to verify variance equality. Effect size was determined using the Rank-Biserial correlation. To ascertain whether any correlation existed between lyso-Gb1 of each patient and their contemporary clinical and laboratory data (age, time on treatment, Cht activity, hemoglobin, platelet count, IgG levels and GD severity scores), the Spearman correlation analyses were conducted. These analyses were segregated based on Type I and Type III. P-values below 0.05 were considered statistically significant. JASP version 0.16.2.0 (JASP Team, 2022) was used for the statistical analysis.” This was modified in lines 178-189.

  1. Table 1 appears overly detailed for the main text. I suggest moving this detailed table to the supplementary material and presenting aggregated data in the main text. Additionally, including IgG levels in the detailed table would be beneficial.

Answer: The table was relabeled to specify the latest clinical and laboratory assessment of each patient, including the addition of IgG in one of the columns as shown in line 191. The authors emphasize the importance of retaining the table within the text to have also a comparison with Table 2.

  1. If the distribution of lyso-Gb1 is not normal, using median values instead of averages would be more appropriate.

Answer: Thank you for your comments and for bringing this to our attention. Median values were used for the statisticall analysis. We have changed all the mean values in the text by median and IQR as we agree with the reviewer. This was corrected in lines 211, 212, 214, 216, 217 and 218.

  1. In comparing the treated GD1 and GD3 groups, it's important to consider whether the treatment durations were similar, as this could impact lyso-Gb1 levels.

Answer: Thank you for your comment. The average duration of treatment in GD1 was 14 years (range: 4-27 years), while the average treatment duration in the GD3 group was 19 years (range: 16-23 years). This information has been included in lines 194-195. For detailed years of treatment, please refer to Table 1. The lyso-Gb1 levels were still higher in the GD3 group.

  1. The significant correlation found between lyso-Gb1 and age in the treated GD1 group warrants further analysis. Adjusting for variables like disease severity, treatment duration, adherence to treatment, etc, is needed to validate this finding. A similar adjustment is necessary for understanding the relationship between IgG levels and lyso-Gb1.

Answer: Thank you for your comments. Our article is exploratory in nature, and will highlight this suggesting that further studies with larger samples be conducted to perform more complex analyses. Additionally, we pointed out as limitations of the study the fact that we did not conduct more complex analyses due to the sample size. This perspective was included in lines 355-358. This was also corrected in the title of the manuscript – line 4.

  1. The absence of correlations in the GD3 treated group could be due to the small sample size (only three cases). This limitation should be acknowledged.

Answer: Thank you for your comment. This was the reason why we couldn’t find any correlations. This information has been added in lines 225-226.

  1. Section E does not contribute significantly to the study of the existing literature. I recommend considering its removal for a more focused and impactful paper.

Answer: Thank you for your comment. Section E has been relocated to the supplementary material, as it was deemed distracting from the main focus of the paper.

  1. Discussion:
    • "We believe that the CSF measurement of lyso-Gb1 could be used as a predictor of the development of PD in GD1 patients."- what is the basis of this belief? Is it something that can be learned from your data? The samples were taken from a PD prodromal study- was there any association between the lyso-Gb1 levels and prodromal findings?

Answer: Thank you for your comment. This same cohort was the same published in a study for non-motor symptoms (NMD) of Parkinson’s Disease. The only patients that  who screened positive for more than one non-motor symptom of Parkinson’s disease were patient’s 7, 13 and 19 [1]. However, these patients didn’t have the highest values of lyso-gb1 in CSF in our present cohort. Therefore, we agree with the reviewer we have added this information and we have modified the sentence in lines 302-311.We have also added the number of NMS on Table 2.

Minor Points:

  1. The gene name should be updated to GBA1 as per recent nomenclature changes.

Answer: Thank you for bringing this to our attention. Corrections have been made in lines 76, 275, and 347 to address the mentioned issue.

  1. References 8 and 9 are duplicated; this should be corrected.

Answer: Thank you for bringing this to our attention. This was corrected in the reference list.

Reference. 1.    Wilke, M.V.M.B.; Poswar, F.; Borelli, W.V.; Michelin Tirelli, K.; Randon, D.N.; Lopes, F.F.; Pasetto, F.B.; Sebastião, F.M.; Iop, G.D.; Faqueti, L.; et al. Follow-up of Pre-Motor Symptoms of Parkinson’s Disease in Adult Patients with Gaucher Disease Type 1 and Analysis of Their Lysosomal Enzyme Profiles in the CSF. Orphanet J. Rare Dis. 2023, 18, 309, doi:10.1186/s13023-023-02875-3.

Reviewer 2 Report

Comments and Suggestions for Authors

Dear Author,

I have reviewed your manuscript titled "A Brazilian Rare Disease Center's Experience with Glucosylsphingosine (lyso-Gb1) in Patients with Gaucher Disease: Novel Correlation with IgG Levels and Biomarker Measurement in CSF" and have some suggestions for your consideration during revision:

While the study provides valuable insights, I recommend the following revisions to strengthen the manuscript:

1. In Table 1, please confirm the total number of patients categorized as having Type 1 Gaucher disease, as there seems to be one patient missing.

2. The use of metachromatic leukodystrophy (MLD) patients as controls for cerebrospinal fluid lyso-Gb1 measurements in this study should be justified, given samples from healthy individuals were likely available. Please expand on the rationale behind selecting this patient group over normal controls. (Page 11, Line 241)

3. In Figure 1A, there is a significant difference in lyso-Gb1 levels between the treated and untreated groups. But did the authors calculate an effect size between the two groups to quantify the magnitude of the treatment effect?

4. A correlation between lyso-Gb1 levels and IgG and age was found in GD1 patients. Does this correlation also exist in GD2 and GD3 subtypes? If not, what may be the reason for this discrepancy?

5. The study reports increased CSF lyso-Gb1 levels in 5 GD1 patients. Does this increase imply neurological involvement in GD1 as well? If so, did the authors discuss related mechanisms for this phenomenon?

6. Case reports of 3 treatment-naive GD patients are provided in the results. Was there any further follow-up of these specific patients after treatment initiation to observe trends in biochemical markers over time?

Please let me know if you need any clarification or have additional questions!

I hope you find these suggestions helpful for revising and polishing your manuscript. Please do not hesitate to contact me if you have any questions. I look forward to reading the revised draft.

Comments on the Quality of English Language

Minor editing of English language required

Author Response

  1. In Table 1, please confirm the total number of patients categorized as having Type 1 Gaucher disease, as there seems to be one patient missing.

Answer: Thank you for bringing this to our attention. The number was corrected in the text in line 191.

  1. The use of metachromatic leukodystrophy (MLD) patients as controls for cerebrospinal fluid lyso-Gb1 measurements in this study should be justified, given samples from healthy individuals were likely available. Please expand on the rationale behind selecting this patient group over normal controls. (Page 11, Line 241)

Answer: Thank you for your comment. CSF from healthy individuals was unavailable at our center during the study, as lumbar punctures were solely conducted in emergency settings in these individuals, and they are not available for research purposes.  However, CSF for MLD was available for research purposes. This explanation was added in lines 168-170.

  1. In Figure 1A, there is a significant difference in lyso-Gb1 levels between the treated and untreated groups. But did the authors calculate an effect size between the two groups to quantify the magnitude of the treatment effect?

Answer: Thank you for your question. We calculated the effect size using Rank-Biserial correlation, with a correlation of -0.928 (95% CI -0.972 to -0.816). This information was now added in lines 212-213.

  1. A correlation between lyso-Gb1 levels and IgG and age was found in GD1 patients. Does this correlation also exist in GD2 and GD3 subtypes? If not, what may be the reason for this discrepancy?

Answer: Thank you for your comment, this is a limitation of our study. A sentence addressing this issue was added “In the GD3 treated group, these correlations were not found due to a smaller sample size.” In lines 225-226.

  1. The study reports increased CSF lyso-Gb1 levels in 5 GD1 patients. Does this increase imply neurological involvement in GD1 as well? If so, did the authors discuss related mechanisms for this phenomenon?

Answer: Thank you for your question. we have included a concise statement regarding the existing knowledge surrounding the link between Gaucher and Parkinson’s disease, highlighting the hypothesis of abnormal GCase contributing to aberrant lipid metabolism and lyso-Gb1 accumulation. Additionally, we have incorporated the latest study by Smith and Schapira (2022) as a reference (https://doi.org/10.3390/cells11081261). Notably, this cohort aligns with a prior investigation on non-motor symptoms of Parkinson’s disease, revealing that patients 7, 13, and 19 were the only ones screening positive for multiple non-motor symptoms [1]. Intriguingly, these individuals did not exhibit the highest lyso-Gb1 values in the CSF within our current cohort. We concur with the reviewer's suggestion, integrating this information and adjusting the sentence in lines 302-312. Furthermore, we have updated Table 2 to include the number of non-motor symptoms (NMS).

  1. Case reports of 3 treatment-naive GD patients are provided in the results. Was there any further follow-up of these specific patients after treatment initiation to observe trends in biochemical markers over time?

Thank you for your comment. Regrettably, the patient with GD2 has passed away, as detailed in the case vignette. Currently, our group continues to follow up with the patients diagnosed with GD1 and GD3, this was added in the supplementary materials.

Reviewer 3 Report

Comments and Suggestions for Authors

This research article showed a correlation of glucosyl-sphingosine (lyso-Gb1) with IgG levels in Gaucher patients. This correlation was claimed as a new correlation. However, there was no in-depth discussion on this point.

Generally, the manuscript was well described in the methodology, demographic, and clinical characteristics of the patients. The findings were not novel, but the article showed the importance of treatments and the changes in biomarkers. Therefore, this article has merit to be published.

There are only minor points in the figures that have to be improved for better understanding.

 1. The graphic size and font size should be consistent. I may suggest that the color for Type I and Type II GD (Figure 1B) should be different from Figure 1A, and the correlation Figure 1C-1E may use the color accordingly. It may help to promptly understand the results without reading the caption. As well as Figure 3, it need to be modified.

2. Figure 3. The vertical dotted line denotes the start of treatment. The results in the text were different from the figure

Line 271-272 and Figure 3 A: Three months after treatment, the level of lyso-Gb-1 decreased to 91 nmol/L, but the value of 91 was shown at about ten months after treatment. Please check Lines 288-298, 308-309, and the figure 3B-3C.

Comments on the Quality of English Language

No additional comment. 

Author Response

There are only minor points in the figures that have to be improved for better understanding.

  1. The graphic size and font size should be consistent. I may suggest that the color for Type I and Type II GD (Figure 1B) should be different from Figure 1A, and the correlation Figure 1C-1E may use the color accordingly. It may help to promptly understand the results without reading the caption. As well as Figure 3, it need to be modified.

Answer: Thank you for your comments. The images have been corrected, and green was used to define continuity between figures 1A and B, and C, D, E.

  1. Figure 3. The vertical dotted line denotes the start of treatment. The results in the text were different from the figure

Answer: Thank you for bringing this to our attention. This was modified in all the figures, in the text corrections were made in the text that is now part of the supplementary materials and highlighted.

  1. Line 271-272 and Figure 3 A: Three months after treatment, the level of lyso-Gb-1 decreased to 91 nmol/L, but the value of 91 was shown at about ten months after treatment. Please check Lines 288-298, 308-309, and the figure 3B-3C.

Answer: Thank you for bringing this to our attention. This was modified in all the figures, in the text corrections were made in the text that is now part of the supplementary materials and highlighted.

Round 2

Reviewer 1 Report

Comments and Suggestions for Authors

Thank you for addressing the comments.

Some additional points - 

Methods:

"A convenience sampling strategy was employed for data collection"- why not all data available? 

From the methods, it is unclear which time-point was used for the Mann-Whitney and Spearman studies. The first? Last? Other? If all time points were used, the analysis could be very biased. Please consult a biostatistician who could explain the problem and suggest a solution. 

Lyso 1 in row 179 should be changed. 

" Type I and Type III " before you used- GD1 and GD3. Suggest being consistent.

Why were patients untreated- physician decision? Patient decision? Availability? 

Discussion-

"In our cohort, we exclusively evaluated patients who were already undergoing treatment to assess age associations." - but, as shown in Figure 1- age was also associated with DS3; thus, although you were using data from treated patients, it could still be that the association is actually with disease severity and not with age.  

Comments on the Quality of English Language

The paper will benefit from English Medical Editing. 

Author Response

REVIEWER #1:

Some additional points - 

Methods:

  • "A convenience sampling strategy was employed for data collection"- why not all data available? 

Answer: Thank you for your comment. We used the term "convenience sampling strategy" to refer to a non-random sampling method where data were collected from individuals available to the researcher. Part of the study period occurred during the SARS-CoV-2 pandemic when many patients had online appointments and were unable to come to the hospital for sample collection. This information has been included in lines 291-292.

2)From the methods, it is unclear which time-point was used for the Mann-Whitney and Spearman studies. The first? Last? Other? If all time points were used, the analysis could be very biased. Please consult a biostatistician who could explain the problem and suggest a solution. 

Answer: Thank you for your comment. The previous version of the manuscript contained an error regarding the dates when the lyso-Gb1 samples were collected; they ranged from January 2018 to March 2023, not from January 2012 as stated. This has been corrected in line 293. Regarding the statistical methods, all available measurements or all patients were included. As reported in the results section, an average of 1.4 measurements per patient was available. As there hasn't been a change in treatment or dosage of the ERT, even during the pandemic when many patients were able to receive their medication in their local hospitals, we believe there aren't many biases in the analysis. This study is exploratory in nature, seeking to uncover new insights into the effects of lyso-Gb1 samples over time. The caveats of our study are highlighted in lines 279-282.

  • Lyso 1 in row 179 should be changed. 

Answer: Thank you for your comment. This has been corrected in line 368.

  • " Type I and Type III " before you used- GD1 and GD3. Suggest being consistent.

Answer: Thank you for your comment. This has been corrected in line 137, 163, 165, 369.

  • Why were patients untreated- physician decision? Patient decision? Availability? 

Answer: Thank you for your comment. These were patients referred directly from primary care facilities to our hospital, specifically to the Gaucher Disease Referral Center for diagnosis and management. Measurements were taken pre- and post-treatment in these patients. This information has been added in lines 152-153.

Discussion-

"In our cohort, we exclusively evaluated patients who were already undergoing treatment to assess age associations." - but, as shown in Figure 1- age was also associated with DS3; thus, although you were using data from treated patients, it could still be that the association is actually with disease severity and not with age.  

Answer: Thank you for your comment. This is very important, and we appreciate you bringing this to our attention. It has been rephrased and included in the discussion in lines 193-201. “In our cohort, we were only able to assess the inverted correlation of Lyso-Gb1 values and age in GD1 treated patients, as shown in Figure 1C. Interestingly, age and disease severity, represented by Disease Severity Score (DS3) and Zimran Severity Score Index (SSI) had a direct statistically significant correlation as shown in Figure 2, even though disease severity and Lyso-Gb1 measurements did not show an association in the present study. Our findings also did not reveal any association between the levels of Lyso-Gb1 and the duration of treatment, as illustrated in Figure 2. We hypothesize that this lack of association might be attributed not only to a small sample size but also to some patients not adhering to treatment, as shown in Table 1.”

  • The paper will benefit from English Medical Editing. 

Answer: Thank you for your comment. English was reviewed again in the final version after the corrections.

Reviewer 2 Report

Comments and Suggestions for Authors

Dear authors,

I thoroughly enjoyed reading your insightful manuscript exploring the utility of lyso-Gb1 measurements in patients with Gaucher disease. You have conducted a well-designed study generating compelling findings regarding correlations with IgG and potential CSF involvement. I believe your work makes a strong contribution, but wanted to offer some friendly suggestions that I feel could further strengthen your publication.

1. For the CSF lyso-Gb1 levels, since MLD patient samples were used for comparison, the authors could comment on whether lyso-Gb1 levels are expected to be different in MLD patients compared to healthy controls. This would further justify the use of MLD patients as the control group.

2. Regarding the correlation between lyso-Gb1 and IgG/age in GD1 but not GD3 patients, the authors could elaborate more on possible reasons for this discrepancy besides smaller GD3 sample size. For example, are there any pathophysiologic differences between subtypes that may lead to a lack of correlation in GD3?

3. The suggested connection between CSF lyso-Gb1 and neurological involvement in GD1 is interesting but still speculative. The authors could discuss any evidence still needed to confirm this hypothesis for future studies.

Please know these suggestions come from a genuine interest in your impressive work. I aim only to provide constructive feedback toward improving an already strong manuscript. Your study represents an important clinical advancement, and I hope you will consider these ideas prior to publication. Please feel free to contact me if you wish to discuss further. I look forward to seeing the final published piece!

Best regards

Comments on the Quality of English Language

Minor editing of English language required

Author Response

REVIEWER #2:

I thoroughly enjoyed reading your insightful manuscript exploring the utility of lyso-Gb1 measurements in patients with Gaucher disease. You have conducted a well-designed study generating compelling findings regarding correlations with IgG and potential CSF involvement. I believe your work makes a strong contribution, but wanted to offer some friendly suggestions that I feel could further strengthen your publication.

  1. For the CSF lyso-Gb1 levels, since MLD patient samples were used for comparison, the authors could comment on whether lyso-Gb1 levels are expected to be different in MLD patients compared to healthy controls. This would further justify the use of MLD patients as the control group.

Answer: Thank you for your comment. This is an interesting topic of debate. In our previous study, we measured chitotriosidase levels in the same samples, and ChT levels exhibited a higher increase in the MLD group (mean = 142.8 ± 101.2 nmol/h/mL; reference range in plasma = 8.8–132 nmol/h/mL, with no available reference range in CSF) compared to the GD group (mean = 70 ± 51 nmol/h/mL), although they were not statistically different (p=0.08) (PMID: 37784132). The more severe brain involvement in MLD might be responsible for the higher levels of ChT found in this group compared to GD patients. Higher levels of ChT in CSF were also demonstrated in other LSDs with more severe neurological involvement than GD1, such as GM1-gangliosidosis and GM2-gangliosidosis, for example. If we believe that lyso-Gb1 and ChT would go "hand in hand," it would be expected for lyso-Gb1 to also be higher in MLD compared to GD1, which was not what was found in our samples, corroborating that lyso-Gb1 is a disease-specific biomarker for GD. This was added in lines 244-249.

  1. Regarding the correlation between lyso-Gb1 and IgG/age in GD1 but not GD3 patients, the authors could elaborate more on possible reasons for this discrepancy besides smaller GD3 sample size. For example, are there any pathophysiologic differences between subtypes that may lead to a lack of correlation in GD3?

Thank you for your comment. This correlation was not made due to the small sample size (n=3) of patients, with one of them not having an available IgG measurement by the time of the lyso-Gb1 sample collection, as shown in Table 1. This has been corrected in lines 137-138.

  1. The suggested connection between CSF lyso-Gb1 and neurological involvement in GD1 is interesting but still speculative. The authors could discuss any evidence still needed to confirm this hypothesis for future studies.

Thank you for your comment. This has been corrected in lines 319-321.
